# Targeting Redox Signaling Through Exosomal MicroRNA: Insights into Tumor Microenvironment and Precision Oncology

**DOI:** 10.3390/antiox14050501

**Published:** 2025-04-22

**Authors:** Moon Nyeo Park, Myoungchan Kim, Soojin Lee, Sojin Kang, Chi-Hoon Ahn, Trina Ekawati Tallei, Woojin Kim, Bonglee Kim

**Affiliations:** 1College of Korean Medicine, Kyung Hee University, 1-5 Hoegidong, Dongdaemun-gu, Seoul 02447, Republic of Korea; mnpark@khu.ac.kr (M.N.P.); dongoorai@khu.ac.kr (M.K.); lettergr@khu.ac.kr (S.L.); bmb1994@khu.ac.kr (S.K.); ach2565@khu.ac.kr (C.-H.A.); wjkim@khu.ac.kr (W.K.); 2Department of Biology, Faculty of Mathematics and Natural Sciences, Universitas Sam Ratulangi, Manado 95115, Indonesia; trina_tallei@unsrat.ac.id; 3Department of Biology, Faculty of Medicine, Universitas Sam Ratulangi, Manado 95115, Indonesia; 4Korean Medicine-Based Drug Repositioning Cancer Research Center, College of Korean Medicine, Kyung Hee University, Hoegi-dong, Dongdaemun-gu, Seoul 02447, Republic of Korea

**Keywords:** ROS–miRNA–exosome axis, redox signaling, miRNA cargo sorting, extracellular vesicles, phytochemicals

## Abstract

Reactive oxygen species (ROS) play a dual role in cancer progression, acting as both signaling molecules and drivers of oxidative damage. Emerging evidence highlights the intricate interplay between ROS, microRNAs (miRNAs), and exosomes within the tumor microenvironment (TME), forming a regulatory axis that modulates immune responses, angiogenesis, and therapeutic resistance. In particular, oxidative stress not only stimulates exosome biogenesis but also influences the selective packaging of redox-sensitive miRNAs (miR-21, miR-155, and miR-210) via RNA-binding proteins such as hnRNPA2B1 and SYNCRIP. These miRNAs, delivered through exosomes, alter gene expression in recipient cells and promote tumor-supportive phenotypes such as M2 macrophage polarization, CD8^+^ T-cell suppression, and endothelial remodeling. This review systematically explores how this ROS–miRNA–exosome axis orchestrates communication across immune and stromal cell populations under hypoxic and inflammatory conditions. Particular emphasis is placed on the role of NADPH oxidases, hypoxia-inducible factors, and autophagy-related mechanisms in regulating exosomal output. In addition, we analyze the therapeutic relevance of natural products and herbal compounds—such as curcumin, resveratrol, and ginsenosides—which have demonstrated promising capabilities to modulate ROS levels, miRNA expression, and exosome dynamics. We further discuss the clinical potential of leveraging this axis for cancer therapy, including strategies involving mesenchymal stem cell-derived exosomes, ferroptosis regulation, and miRNA-based immune modulation. Incorporating insights from spatial transcriptomics and single-cell analysis, this review provides a mechanistic foundation for the development of exosome-centered, redox-modulating therapeutics. Ultimately, this work aims to guide future research and drug discovery efforts toward integrating herbal medicine and redox biology in the fight against cancer.

## 1. Introduction

### 1.1. Roles of Exosomes, miRNA, and Reactive Oxygen Species (ROS) in the Tumor Microenvironment (TME)

Cancer-associated fibroblasts (CAFs) are a highly heterogeneous stromal cell population in the tumor microenvironment (TME), arising from diverse precursors such as tissue-resident fibroblasts [1,2], stellate cells [3,4], mesenchymal stem cells (MSCs) [5,6], white adipocytes [7,8], epithelial and endothelial cells, and monocytes [9,10,11,12,13,14]. Their activation is orchestrated by a network of tumor-derived factors, including cytokines transforming growth factor beta (TGF-β), platelet-derived growth factor (PDGF), Hepatocyte Growth Factor (HGF), chemokine C-C motif ligand 2 (CCL2), C-X-C motif chemokine ligand 12 (CXCL12), and inflammatory mediators such as Interleukin-1β (IL-1β) and Interleukin 6 (IL-6) [5,15,16,17,18,19]. Notably, reactive oxygen species (ROS) function as critical modulators, inducing fibroblast activation, differentiation, and phenotypic reprogramming through signaling cascades such as p38 mitogen-activated protein kinases (p38 MAPKs), epithelial–mesenchymal transitions (EMTs), and EndMT [12,20,21,22]. In parallel, tumor-derived exosomes serve as pivotal vehicles in the horizontal transfer of oncogenic cues that promote CAF conversion. These extracellular vesicles carry proteins, lipids, and nucleic acids, including complexes like CD44v6/C1QBP that activate hepatic stellate cells (HSCs) and enhance Extracellular Matrix (ECM) remodeling and metastatic potential [23]. Exosomal communication works synergistically with ROS to drive inflammation, matrix stiffening, and angiogenesis within the TME. Furthermore, CAFs can respond to damage-associated molecular patterns (DAMPs) released by necrotic tumor cells, activating the NLR family pyrin domain containing 3 (NLRP3) inflammasome and propagating tumor-promoting inflammation [24]. Epigenetic regulators such as leukemia inhibitory factor (LIF) reinforce CAF pro-invasive activity by modulating signal transducer and activator of transcription 3 (STAT3) acetylation and Janus kinase 1 (JAK1) signaling [25,26]. Together, these findings underscore that ROS and exosomes act as master regulators in the multi-lineage differentiation and sustained activation of CAFs, positioning them as crucial therapeutic targets in the context of cancer progression and TME remodeling [27,28,29,30,31,32]. In addition to CAFs, dendritic cells (DCs) also contribute significantly to the modulation of the TME through ROS-mediated exosomal mechanisms. Notably, natural compounds promote calreticulin (CRT) and heat-shock protein (HSP) expression in DCs, enhancing their capacity to prime T cells, while matrine upregulates co-stimulatory molecules, including tumor necrosis factor-α (TNF-α), Interleukin 12 (IL-12), cluster of differentiation 80 (CD80), and CD86, further amplifying antitumor immunity [33].

### 1.2. Key Redox-Associated Signaling Pathways Regulated by Exosomal miRNAs

Exosomes are nanoscale extracellular vesicles, typically ranging from 30 to 100 nm in diameter, that have gained significant attention for their role in cell–cell communication within the TME [34]. These vesicles transport various bioactive molecules, including microRNAs (miRNAs), long noncoding RNAs (lncRNAs), and messenger RNAs (mRNAs), influencing multiple aspects of tumor progression, immune evasion, and therapeutic resistance [35,36]. Given their ability to shuttle regulatory molecules between cells, exosomes are emerging as key mediators of tumor-host interactions and potential targets for cancer diagnosis and treatment [37]. miRNAs are small, noncoding RNAs (19–22 nucleotides) that play a central role in post-transcriptional gene regulation. Within the TME, exosomal miRNAs exhibit remarkable stability due to their encapsulation, protecting them from degradation by RNases [38]. Upon uptake by recipient cells, these miRNAs can modulate gene expression and interfere with intracellular signaling pathways, thereby influencing tumor cell proliferation, metastasis, and resistance to therapy [39]. Notably, tumor-derived exosomes (TEXs) closely reflect the molecular composition of their originating cancer cells [40], providing an accurate representation of the TME. Through their cargo, exosomes can reprogram recipient cells, contributing to angiogenesis, immune suppression, and metastatic spread [41,42]. Notably, one of the most critical aspects of the exosome-ROS-miRNA network is its ability to regulate the tumor immune microenvironment. In particular, tumor-associated macrophages (TAMs), which predominantly exhibit an M2-like immunosuppressive phenotype, are key players in tumor immune evasion. Recent studies suggest that exosomal miRNAs can modulate macrophage polarization by altering oxidative stress levels and redox signaling [43]. For example, exosomal miR-155-5p has been shown to promote immune evasion by suppressing antitumor immunity through the modulation of Programmed Cell Death Ligand 1 (PD-L1) expression in macrophages [44]. Similarly, exosomal miR-21 and miR-146a can induce macrophage reprogramming towards an M2 phenotype, reinforcing an immunosuppressive TME [45]. In addition to macrophages, exosomal miRNAs influence fibroblast activation and immune cell function through redox-sensitive pathways. Transforming growth factor-beta 1 (TGFβ1)-induced oxidative stress has been implicated in transforming normal fibroblasts into CAFs [46], which secrete additional exosomes containing miRNAs that further modulate the redox balance and tumor progression [47]. The impact of exosomal miRNAs on tumor redox homeostasis is largely mediated through key antioxidant and oxidative stress signaling pathways, including the nuclear factor erythroid 2-related factor 2 (Nrf2)/Kelch-like ECH-associated protein 1(Keap1) pathway, and the glutathione peroxidase 4 (GPX4)-related ferroptosis regulation and TGFβ1-ROS axis. The transcription factor Nrf2 is a major regulator of antioxidant defenses and cellular redox homeostasis. Exosomal miRNAs, such as miR-200c, have been shown to suppress Nrf2 activation, leading to reduced antioxidant capacity and increased oxidative stress in recipient cells, thereby sensitizing them to ferroptosis [48]. GPX4 is a crucial enzyme that protects cells from lipid peroxidation and ferroptosis. Exosomal miR-23a-3p has been reported to inhibit GPX4 expression, promoting ROS accumulation and iron-dependent cell death, which can influence cancer progression [49]. TGFβ1 plays a dual role in tumor progression by promoting both oxidative stress and immune suppression. Exosomal miRNAs, such as miR-34a and miR-1246, can activate TGFβ1 signaling, leading to increased ROS production and fibroblast-mediated tumor progression [50,51]. In light of this, miRNA-loaded exosomes derived from ROS-modulating natural compounds offer additional layers of regulation, especially in the context of dendritic cell function enhancement, as observed with matrine [52]. This section aims to dissect how specific exosomal miRNAs orchestrate redox-responsive signaling pathways and reprogram the immune landscape, offering mechanistic insights into their potential as therapeutic gatekeepers in oxidative stress-driven cancers.

### 1.3. Functional Genomics and Single-Cell Approaches in Understanding the Exosome-ROS-miRNA Axis

The intricate interplay between exosomes, miRNAs, and ROS within the TME has led to increasing use of advanced genomic and transcriptomic technologies to better understand their roles. Functional genomics approaches, such as clustered regularly interspaced short palindrome repeat (CRISPR)-based screens and high-throughput sequencing, have been instrumental in identifying key exosomal miRNAs that contribute to redox regulation in various cancer types [18]. Furthermore, cutting-edge tools like single-cell RNA sequencing (scRNA-seq) and spatial transcriptomics have allowed for cell-type-specific resolution of exosome-related signaling and redox balance across diverse immune and stromal populations in the TME [53,54]. These technologies have revealed the heterogeneous nature of exosome-mediated communication and how it dynamically shapes redox states, immune phenotypes, and tumor progression at the single-cell level. Long noncoding RNAs (lncRNAs), once regarded as transcriptional noise, are now recognized for their regulatory significance, with over twice as many lncRNA genes compared to protein-coding genes in the human genome. Despite being less studied than miRNAs, accumulating evidence indicates that lncRNAs participate in numerous physiological and pathological processes, including cell cycle control, differentiation, tumorigenesis, epigenetic modification, and chromatin remodeling [55,56]. Exosomes facilitate the intercellular exchange of miRNAs by enclosing them in lipid bilayers that protect them from degradation by RNases. The content of circulating miRNAs differs significantly from tissue-derived miRNAs due to variations in the mechanisms of release, such as passive leakage from necrotic cells or active secretion via exosomes. Notably, circulating miRNA profiles are consistent between genders, suggesting that their patterns are not sex-dependent [57,58,59,60]. Once internalized by recipient cells, exosomal miRNAs can regulate gene expression or directly interact with intracellular proteins, modifying their biological function [37]. ROS, generated by cytokines like TGF-β1, not only play a role in oxidative stress but also contribute to the transformation of normal fibroblasts (NFs) into cancer-associated fibroblasts (CAFs) [61,62]. Accumulated ROS can alter miRNA expression profiles, contributing to tumorigenesis. Given that both ROS and miRNAs are differentially expressed in cancer, their interplay may offer novel therapeutic targets [30]. For instance, miRNAs have been shown to regulate intracellular ROS production through several redox-related pathways, including Nrf2/Keap1, mitochondrial antioxidant enzymes, and the SOD/catalase axis [48]. Recent findings also highlight the ability of exosomes to mediate cross-species communication, delivering therapeutic RNA payloads including miRNAs and lncRNAs across biological kingdoms [63]. Cancer cells generally present elevated ROS levels due to metabolic reprogramming and dysregulated signaling. While high ROS levels can promote tumor progression, excessive ROS may also induce cell death via apoptosis or necrosis [64,65]. Exploiting this vulnerability, several chemotherapeutic and radiotherapeutic agents aim to induce ROS-mediated cytotoxicity in cancer cells [66]. A critical ROS-related cell death mechanism in cancer is ferroptosis, which is closely linked to iron metabolism. Iron overload induces ROS production via the Fenton reaction, leading to lipid peroxidation and ferroptotic cell death [67]. Transferrin-bound Fe^3+^ enters cells through transferrin receptors (TFRCs) and is reduced to Fe^2+^ in endosomes. Divalent metal transporter 1 (DMT1) then transports Fe^2+^ to the cytoplasm, where it amplifies ROS production [68]. Exosomes have been shown to regulate this process; for example, exosomal lncRNA nuclear-enriched transcript 1 downregulates miR-9-5p, thereby increasing TFRC expression and promoting ferroptosis, as reflected by elevated ROS and malondialdehyde (MDA), and decreased GPX4 and glutathione (GSH) [69]. Conversely, exosomal miR-23a-3p suppresses DMT1 expression, inhibiting ferroptosis by increasing GSH and reducing Fe^2+^ and MDA levels [70]. TAMs consist of M1 (pro-inflammatory, antitumor) and M2 (anti-inflammatory, pro-tumor) phenotypes. In most tumor contexts, M2 TAMs dominate and contribute to immune suppression [51]. Reprogramming M2 to M1 macrophages or eliminating M2 cells has become a promising immunotherapeutic strategy [40]. Iron overload can facilitate M1 polarization by upregulating M1 markers, enhancing glycolysis, and increasing ROS and p53 acetylation [71]. For instance, downregulation of Apolipoprotein C1 (APOC1) has been shown to increase iron and ROS levels in TAMs, triggering ferroptosis and promoting M2-to-M1 repolarization, which impairs tumor growth. Exosomes also modulate the effectiveness of immunotherapy [72,73]. Ferroptosis has been demonstrated to be immunogenic, both in vitro and in vivo, and may help overcome resistance to cancer immunotherapy [74,75]. Its impact is influenced by interactions with immune cells, cancer cells, and the broader TME [76]. Engineered exosomes targeting ferroptosis pathways are being investigated as therapeutic tools to inhibit tumor growth [70]. For example, exosomes have been shown to activate the SIRT1/Nrf2/ Heme oxygenase-1 (HO-1) signaling axis, increasing GPX4 and GSH while reducing ROS, MDA, and Fe^2+^ levels, thereby suppressing ferroptosis [77]. Additionally, exosome-mediated delivery of glutamic-oxaloacetic transaminase 1 (GOT1) has been shown to upregulate C-C motif chemokine receptor 2 (CCR2) and stimulate Nrf2–HO-1 signaling, promoting GPX4 expression and reducing ROS, MDA, and Fe^2+^, and ultimately enhancing tumor cell migration and invasion [78,79]. This section highlights how recent advances in single-cell and spatial omics technologies are enabling a deeper understanding of redox-regulated miRNA–exosome interactions, and provides a conceptual framework for developing precision-targeted cancer therapies based on these insights. Moreover, DCs are increasingly recognized as contributors to redox and immune modulation in the TME through their exosome secretion. DC-derived exosomes enriched in TGF-β, IL-10, PD-L1, and immunoregulatory miRNAs such as miR-155 and miR-146a have been shown to suppress CD8+ T cell responses and promote a tolerogenic microenvironment. These findings further support the role of immune cell-derived exosomes in orchestrating ROS-linked immunosuppression and cellular reprogramming within the TME (Figure 1).

## 2. The Influence of ROS and Exosome Biogenesis and miRNA Loading

### 2.1. NOX1-Driven ROS Production and Its Role in Tumor Progression and Immune Modulation

Hypoxia and inflammation, both of which are common in the TME, upregulate ROS production and activate transcription factors such as hypoxia-inducible factor-1α (HIF-1α) and nuclear factor kappa-light-chain-enhancer of activated B cells (NF-κB), which have been shown to stimulate exosome biogenesis pathways [80]. ROS, generated as metabolic byproducts or via dedicated enzymatic systems, serve both signaling and pathological roles in cancer. Their involvement in exosome formation and release has become increasingly evident. Low to moderate levels of ROS can inhibit lysosomal degradation of multivesicular bodies (MVBs), thereby promoting the release of exosomes [81]. Conversely, excessive oxidative stress may stimulate autophagy, leading to enhanced MVB degradation and reduced exosome output. This dual role of ROS indicates a dose-dependent regulation of exosome biogenesis. Exosomes contain a wide range of molecular components, including tetraspanins (CD9, CD63, CD81), endosomal sorting complex required for transport (ESCRT) proteins (Alix, TSG101), heat-shock proteins, integrins, and various lipid molecules such as sphingomyelin, ceramides, and cholesterol [82,83]. These molecules contribute to membrane stability, cargo sorting, and cell–cell signaling, and are sensitive to changes in the oxidative status of the cell [80]. Among the key contributors to ROS generation are the NADPH oxidase (NOX) enzymes, particularly NOX1, which is often upregulated in various cancer types. NOX1-derived ROS production has been shown to increase the formation and release of exosomes from tumor cells [84]. NADPH oxidase 1 (NOX1), located on the X chromosome, is a membrane-bound enzyme that plays a central role in the regulated production of ROS. NOX1 has been implicated in a variety of pathological processes, including vascular dysfunction, tumor angiogenesis, and fibrotic disease [85]. Recent studies have demonstrated that overexpression of DEAD-box helicase 19A (DDX19A) enhances NOX1-mediated ROS production, promoting migration and invasion of cervical squamous cell carcinoma cells. Moreover, elevated NOX1 expression correlates with poor overall survival in patients, underscoring its oncogenic potential [86]. Within the TME, TAMs are major players in tumor progression. These innate immune cells constitute approximately 30–50% of the total cellular population in many tumors, including cervical cancer, and are involved in tumor cell invasion, angiogenesis, and immunosuppression [87,88,89]. TAMs are broadly classified into two phenotypes. M1-like TAMs, which secrete pro-inflammatory cytokines and exhibit tumor-suppressive functions. M2-like TAMs, which are characterized by low antigen presentation and support tumor growth, metastasis, and immune evasion [90,91]. During tumor development, macrophages can undergo phenotypic reprogramming from M1-like to M2-like states in response to local signals within the TME, including oxidative stress and tumor-derived exosomes [92]. The connection between NOX1 activity and extracellular vehicles (EVs) has also been explored in recent literature. For instance, a study by Žėkas et al. reported that blood-derived EVs from myocardial infarction patients exhibit increased oxidative activity, suggesting a role for ROS in systemic EV function [93]. Similarly, Renato et al. found that platelet-derived EVs express NOX1, produce superoxide, and modulate platelet responses, supporting the hypothesis that EVs can carry functional ROS-generating enzymes like NOX1 and contribute to redox signaling in various contexts [94]. These findings support the growing recognition that exosomal NOX1 may serve as both a biomarker and a functional mediator of disease progression. In cancer, tumor cell-derived exosomes enriched with NOX1 have been shown to influence recipient immune cells, especially macrophages, by promoting M2 polarization and enhancing a pro-tumorigenic, immunosuppressive environment. This section underscores how ROS production, particularly via NOX1 activation, not only alters exosome biogenesis but also orchestrates immunosuppressive remodeling within the TME, offering insight into redox-targeted strategies for future therapeutic development.

### 2.2. Cellular Source of ROS and Exosome-Linked Immune Modulation

Among the immune cells involved in tumor inflammation, tumor-associated neutrophils (TANs) play a particularly nuanced role. Chronic inflammation is a hallmark of cancer, and nearly 25% of cancers are linked to persistent inflammation and infection [95]. Neutrophils are highly abundant in various solid tumors, including melanoma, head and neck squamous cell carcinoma (HNSCC), renal carcinoma, and bronchoalveolar carcinoma and elevated neutrophil levels in both peripheral blood and tumors have been associated with poor prognosis and reduced recurrence-free survival [96,97,98,99]. Despite their immunological role, TANs exhibit functional plasticity and can polarize into either anti-tumorigenic N1 or pro-tumorigenic N2 phenotypes depending on tumor type, location, and progression stage. This dual nature complicates therapeutic targeting, as a high neutrophil-to-lymphocyte ratio (NLR) may correlate with poor prognosis, but does not fully reflect neutrophil function in the TME. Several neutrophil-targeting agents, such as aspirin (inhibiting prostaglandin synthesis) and CCX168 (blocking the complement C5a receptor, C5aR), have shown potential in treating inflammatory disorders and certain cancers by reducing neutrophil recruitment and activation [100,101,102]. However, selectively modulating pro-tumoral N2 TANs remains a major challenge due to their complex phenotypic diversity and signaling context. Importantly, neutrophil-derived exosomes (NDEs) have emerged as key mediators of neutrophil-driven tumor progression. In other inflammatory conditions like asthma, NDEs have been shown to promote airway smooth muscle (ASM) cell proliferation and alter macrophage and dendritic cell activation, thereby illustrating their broad immunomodulatory capacity. In the tumor context, these exosomes contain immunosuppressive molecules such as TGF-β, IL-10, IL-6, prostaglandin E2 (PGE2), and Foxp3, which can suppress cytotoxic immune responses by disarming CD4^+^ T-cells, NK cells, and macrophages [103,104]. Moreover, tumor-derived exosomes (TEXs) can actively polarize neutrophils toward the N2 phenotype and promote autophagy via the high mobility group protein 1 (HMG-1)/toll-like receptor 4 (TLR4)/NF-κB signaling pathway [105], facilitating immune escape and tumor survival. Thus, neutrophils not only act as a cellular source of ROS and inflammatory mediators but also participate in exosome-mediated reprogramming of the immune microenvironment. These findings reinforce the interconnected roles of ROS, neutrophils, and exosomes in shaping tumor immunity and suggest that precise targeting of TAN-derived or tumor-derived exosomes may offer novel therapeutic opportunities in immuno-oncology [105]. Extending the concept of EV-mediated immune evasion, tumor endothelial cells (TECs), a specialized vascular component of the TME, have recently been shown to actively release extracellular vesicles (TEVs) enriched in mTOR. These TEVs promote systemic immune suppression via the G-CSF pathway, upregulate immune checkpoint molecules such as PD-L1 and LAG3 on T cells, and contribute to redox dysregulation and metastatic spread [106]. Multiple cell types within the TME, including TAMs, CAFs, neutrophils, DCs, TECs, and tumor cells themselves, contribute to ROS production. Each of these cells may release exosomes containing ROS-modulating components or redox-sensitive miRNAs. For instance, CAFs can release exosomal miR-522, which suppresses ferroptosis in gastric cancer by inhibiting arachidonate 15-lipoxygenase (ALOX15), a key lipid-ROS generator [107]. Neutrophils, through NOX-mediated ROS production, can also induce immunosuppression by releasing exosomes that modulate T-cell function via PD-L1 expression and TGF-β-dependent polarization [84,108]. Similarly, dendritic cell-derived exosomes are capable of delivering ROS-related signals to T cells, modulating antigen presentation and promoting immune evasion [91,109]. Exosomes from tumor cells and their surrounding stromal cells act as messengers that transfer ROS-generating machinery (such as NOX1) and redox-sensitive molecules, thereby remodeling the microenvironment in favor of tumor survival and metastasis. The NOX1–ROS–exosome axis is now considered a critical mechanism for cross-talk between tumor cells and immune components, influencing not only redox balance but also TAM polarization, neutrophil behavior, and CD8^+^ T cell suppression [84,110,111]. By highlighting the intricate interplay between diverse immune cell-derived exosomes and oxidative signaling, this section seeks to define novel immunological checkpoints that could be leveraged for therapeutic intervention in redox-dysregulated tumors. To provide a comprehensive overview of how each stromal and immune cell type contributes to ROS regulation, exosomal dynamics, and miRNA-mediated signaling in the tumor microenvironment, we have summarized their distinct roles and molecular interactions in Table 1. This cell-type-specific mapping underscores the coordinated but heterogeneous nature of the ROS–miRNA–exosome axis in cancer biology.

### 2.3. ROS and Selective miRNA Packing into Exosome

Notably, ROS also influence the selective sorting of miRNAs into exosomes. Under oxidative stress, cancer cells exhibit changes in miRNA expression and packaging machinery, leading to preferential inclusion of redox-sensitive miRNAs such as miR-21, miR-155, miR-210, and others [72]. These miRNAs are known to modulate immune response, cell survival, angiogenesis, and resistance to therapy. Interestingly, extracellular vesicle (EV)-associated miRNAs (EV-miRNAs) are more stable and biologically active than their free counterparts, as they are selectively packaged into vesicles that shield them from extracellular RNases. This selective packaging is not a passive process; instead, it is tightly regulated by short RNA sequence motifs known as EXO-motifs, which are enriched in exosomal miRNAs and guide their loading by interacting with specific RNA-binding proteins (RBPs) such as hnRNPA2B1 and SYNCRIP [114]. These RBPs facilitate miRNA recognition and transport into vesicles via ESCRT-dependent and independent pathways. ROS can modulate these pathways either directly or via upstream regulatory cascades that impact RBP activity, ESCRT function, or lipid raft integrity [115,116,117,118]. Moreover, EVs are more than passive carriers of cellular waste; they are emerging as sophisticated mediators of intercellular communication, especially in the crosstalk between hypoxia and inflammation. For instance, hypoxic stress and redox imbalance have been shown to affect the miRNA cargo profile in EVs. EVs enriched with miR-517a-3p, derived from trophoblast cells, are known to target immune cells such as T cells and NK cells, modulating their function [119,120]. Similarly, bovine placenta-derived EVs transfer miR-499 to endometrial epithelial cells, suppressing NF-κB activation and dampening inflammatory responses. These examples support the idea that EV-miRNAs are responsive to environmental stimuli such as ROS and hypoxia, and may be tailored to exert specific effects on recipient cells [119]. Another layer of complexity lies in how EV-miRNAs execute their functions. Once internalized, EVs deliver miRNAs into recipient cells where they regulate gene expression either by inhibiting translation or promoting mRNA degradation. This can result in either immunosuppressive or immunostimulatory outcomes depending on the miRNA content and cell type. For instance, hepatocellular carcinoma-derived EV-miRNAs have been shown to facilitate tumor invasion by inducing TGF-β and TGF-β-activated kinase-1 signaling [121], while MSC-derived EVs enriched in miR-125 improve cardiac function through enhancing autophagy flux [122]. These opposing effects underscore the context-dependent role of EV-miRNAs, influenced by both intracellular ROS levels and the extracellular microenvironment. Collectively, these findings highlight that the sorting of miRNAs into EVs is a complex and dynamically regulated process, in which ROS serve as both a trigger and a modulator. Understanding this fine-tuned mechanism may reveal new therapeutic strategies targeting immune modulation, inflammation control, and tumor suppression via engineered EV-miRNA delivery systems [123]. This section emphasizes the importance of deciphering ROS-sensitive miRNA sorting mechanisms in exosomes, which could pave the way for precision-based interventions using miRNA-engineered vesicles tailored to redox-dependent tumor microenvironments. As illustrated in Figure 2, tumor-derived ROS play a central role in driving the selective packaging of oncogenic or immunomodulatory miRNAs into exosomes. These vesicles are subsequently delivered to multiple cell types, including fibroblasts, immune cells, and endothelial cells, where they potentiate processes such as angiogenesis, epithelial–mesenchymal transition (EMT), and immune evasion. The ROS–miRNA–exosome axis thus acts as a self-amplifying circuit that perpetuates tumor progression and therapeutic resistance.

### 2.4. Mechanistic Insights into ROS-Driven Exosome Secretion and Functional Adaptation

Numerous recent studies have highlighted the complex interplay between oxidative stress and exosome biogenesis, offering new insights into the regulation of MVBs dynamics. It is now well recognized that high levels of ROS can activate autophagy, resulting in the degradation of MVBs and consequently reducing exosome release, whereas low oxidative stress prevents lysosomal degradation of MVBs, thereby promoting exosome secretion [81]. Further mechanistic studies revealed that inhibition of lysosomal function by ROS is mediated through calcium signaling pathways. For example, in homocysteine-treated podocytes, endogenous ROS overproduction impairs lysosomal Ca^2+^ release via the transient receptor potential mucolipin 1 (TRPML1) channel, which disrupts lysosome-MVBs fusion, thereby enhancing exosome secretion [124,125]. Additionally, transcriptional regulators such as transcription factor E (TFE), a member of the microphthalmia/transcription factor E (MiT)/TFE transcription factor family, play a pivotal role. Under physiological conditions, TFE3 is phosphorylated by mTOR and retained in the cytosol. However, under oxidative stress (sulforaphane-induced GSH imbalance), mTOR activity is suppressed, resulting in TFE3 dephosphorylation and nuclear translocation. Once in the nucleus, TFE3 drives lysosomal gene expression but also leads to the formation of dysfunctional lysosomes, impairing their degradation capacity and thereby increasing exosome yield [100,126]. In various models, oxidative stress induced by oxygen–glucose deprivation or hydrogen peroxide (H₂O₂) treatment has been shown to enhance exosome secretion. In astrocytes, these stressors increase the release of exosomes enriched in prion protein (PrP), which are subsequently transferred to neurons to improve their survival under hypoxic conditions [127]. Stress conditions also influence miRNA content in exosomes. For instance, miR-126 levels in exosomes derived from endothelial progenitor cells vary depending on the oxidative environment, although they mirror intracellular levels. These exosomal miR-126 molecules, when delivered to brain microvascular endothelial cells under hypoxia/reoxygenation injury, downregulate ROS production via the Phosphatidylinositol 3-kinase (PI3K) signaling pathway, suggesting a protective feedback mechanism [128]. Hypoxia, a hallmark of solid tumors, has emerged as a critical factor influencing exosome production and function. Cancer cells under hypoxic stress release significantly more exosomes, which are often enriched in specific proteins, lipids, and regulatory RNAs such as miRNAs. This process is largely regulated through the activation of HIF-1α. For instance, King et al. demonstrated that HIF-1α is essential for the elevated release of exosomes from cancer cells under hypoxia [129]. Exosomes derived from hypoxic cancer cells show increased levels of miR-210, a hypoxia-responsive miRNA known to promote angiogenesis [129,130]. In multiple myeloma, hypoxia-resistant cell lines secrete more exosomes with significantly higher miR-135 content compared to their normoxic counterparts, promoting endothelial tube formation via downregulation of factor inhibiting hypoxia-inducible factor 1 (FIH1) [131]. Similarly, Kucharzewska et al. reported that exosomes from glioblastoma multiforme (GBM) cells cultured under hypoxia were enriched in matrix metalloprotease-9 (MMP-9), pentraxin 3, IL-8, PDGF, and Plasminogen activator inhibitor-1 (PAI-1), reflecting the hypoxic gene expression of donor cells and GBM tumors [132]. These exosomes markedly enhanced angiogenesis in vitro and in vivo by altering endothelial cell behavior. In leukemia, exosomes released under hypoxia facilitated endothelial tube formation through miR-210 transfer, as shown by Tadokoro et al. [133]. The acidic conditions associated with tumor hypoxia also contribute to elevated exosome biogenesis and uptake. Acidic pH has been shown to promote both exosome release and cellular uptake, further enhancing intercellular communication within the TME [134]. Early studies also identified hypoxia-induced exosomal proteins that drive angiogenesis and metastatic potential [135]. Notably, transcriptionally active HIF-1α has been detected within exosomes derived from nasopharyngeal carcinoma cells, where it contributes to recipient cell activation [136]. Furthermore, adipocyte-derived exosomes released under hypoxia have been shown to promote lipogenesis in recipient 3T3-L1 cells [137]. Beyond their biological function, exosomes are increasingly being explored as nanocarriers for targeted therapies. Due to their size, they exploit the enhanced permeability and retention (EPR) effect to accumulate in tumors, and can evade clearance by the mononuclear phagocyte system (MPS) [138]. Exosomes demonstrate excellent compatibility and ability to cross physiological barriers, including the blood-brain barrier, and can be administered via multiple routes such as intravenous, intranasal, and intracranial delivery [85,139]. Innovative applications include exosome-based delivery systems for photosensitizers, which can generate ROS upon laser exposure in tumor tissues, achieving effective photodynamic therapy with minimal off-target damage [140]. However, challenges remain: heterogeneous secretion capacity among cell types, low drug-loading efficiency, and complex, low-throughput isolation methods limit their large-scale biomedical applications [86]. Several novel cell-engineering approaches have emerged to address these challenges. For example, Wang et al. generated chimeric cells by integrating tumor cell nuclei into activated M1-like macrophages, producing exosomes capable of homing to lymph nodes and xenograft tumors, and enhancing T-cell activation while relieving immunosuppression [103]. Similarly, mesenchymal stem cell-derived exosomes have been shown to support osteochondral regeneration via CD73-mediated adenosine signaling, promoting M2 macrophage infiltration and reducing proinflammatory cytokines like IL-1β and TNF-α [105]. Taken together, these insights underscore how oxidative stress, particularly under hypoxia, modulates exosome secretion, cargo composition, and functional activity across cancer and stromal cells. From autophagy-driven MVB dynamics to HIF-1α-regulated miRNA packaging, ROS serve as a central integrator of exosomal biology in the TME. This section aims to consolidate emerging mechanistic insights into ROS-responsive exosome adaptation and highlights their translational relevance for designing exosome-based delivery strategies and tumor-targeted therapeutics.

## 3. Natural Compounds Modulating the ROS–miRNA Axis: A New Frontier in Exosome-Based Therapeutics

Several natural compounds derived from herbal medicines have shown potential in modulating ROS levels through miRNA regulation. However, few studies have explored their impact on exosome-mediated communication. Table 2 below summarizes key phytochemicals, their known miRNA targets, and current evidence regarding exosome involvement. These findings highlight the need to investigate how natural products may influence exosome biogenesis or cargo selection, potentially leading to novel redox-targeted therapies. Astragaloside IV and ferulic acid, key active components of traditional Chinese medicine, synergistically ameliorated bleomycin-induced pulmonary fibrosis in mice by reducing oxidative stress and modulating the TGF-β1/Smad3 signaling pathway via miR-29b. Although exosome involvement was not investigated, these findings highlight their potential for exosome-based therapeutic research in fibrosis-related disorders [141]. Baicalin, a flavonoid derived from *Scutellaria baicalensis*, exhibits significant anticancer effects in breast and gynecological cancers by regulating oxidative stress and miRNA expression, including upregulation of tumor-suppressive miR-338-3p and let-7c. It enhances ROS production, suppresses NF-κB and Bcl-2 pathways, and promotes apoptosis, although its role in exosome biology has yet to be explored [142]. Berberine, an alkaloid from *Coptis chinensis*, suppresses NF-κB signaling through Set9-mediated methylation of RelA, thereby downregulating miR-21 and its target Bcl-2, which leads to increased ROS generation and apoptosis in multiple myeloma cells. Although its exosomal role is yet to be explored, these findings suggest strong redox and miRNA-modulatory potential suitable for exosome-based therapeutic strategies [143]. BK002, a natural mixture derived from *Achyranthes japonica* and *Melandrium firmum*, has been shown to upregulate tumor-suppressive miRNAs such as miR-192-5p, inducing ROS-mediated apoptosis in prostate cancer cells. Although exosomal pathways have not been explicitly studied, their potent regulation of the ROS–miRNA axis suggests potential for future exosome-focused research [144]. Recent studies have highlighted the ROS-regulating potential of multi-herbal prescriptions in traditional Chinese Medicine. For instance, Bu-Shen-Ning-Xin decoction (BSNXD) was shown to suppress oxidative stress in premature ovarian insufficiency by modulating the circRNA_012284/miR-760-3p/Heparin-binding EGF-like growth factor (HBEGF) axis, leading to ROS attenuation in ovarian granulosa cells [145]. *Cnidium officinale Makino extract* (COM) was shown to induce apoptosis in U937 and U266 cells by increasing ROS levels and triggering ER stress pathways, including C/EBP Homologous Protein (CHOP) and cleaved Poly [ADP]-Ribose Polymerase (PARP) activation. Importantly, this effect was mediated through downregulation of miR-211, a pro-survival miRNA, highlighting the miRNA–ROS interplay in COM-induced cell death [146]. Liu et al. demonstrated that curcumin induces ROS accumulation, which activates the KEAP1/NRF2 pathway and subsequently upregulates miR-34a/b/c in a p53-independent manner. These miRNAs mediate curcumin’s pro-apoptotic, anti-metastatic, and senescence-inducing effects in colorectal cancer models. This study highlights the therapeutic relevance of curcumin in modulating the ROS–miRNA axis, although its exosome-related mechanisms remain to be explored [147]. Recent research by Chen et al. demonstrated that DanShen Decoction (DSD) pretreatment enhanced the expression of miR-93-5p in bone marrow-derived mesenchymal stem cell (BMSC) exosomes, leading to reduced ROS production and cardiomyocyte pyroptosis via the Thioredoxin-interacting protein (TXNIP)/NLRP3/Caspase-1 pathway. These findings highlight the therapeutic promise of DSD in modulating the ROS–miRNA–exosome axis for myocardial protection [148]. *Daemonorops draco* Blume (DD), a traditional Korean medicinal resin, was shown to induce apoptosis in acute myeloid leukemia (AML) cells by upregulating miR-216b, which inhibits c-Jun and triggers ER stress-mediated ROS accumulation. This study highlights DD’s potential as a multi-target anticancer agent through simultaneous modulation of miRNA and redox pathways, although its effects on exosome biology remain to be investigated [149]. Ginsenoside Rg3, a bioactive component of *Panax ginseng*, was shown to inhibit vasculogenic mimicry and stemness in pancreatic adenocarcinoma (PAAD) by enhancing miR-204 expression in cancer cell-derived exosomes. This exosomal miR-204 targets DVL3, leading to suppression of Wnt/β-catenin and PI3K/protein kinase B (AKT) signaling, thereby reducing tumor progression and mimicking angiogenesis. These findings suggest a potential strategy for overcoming resistance to anti-angiogenic therapies [150]. *Leonurus japonicus* Houttuyn was shown to induce ROS-mediated apoptosis in acute myeloid leukemia (AML) cells via suppression of oncogenic miR-19a-3p, which restored phosphatase and tensin homolog (PTEN) expression and inhibited PI3K/AKT signaling. The study highlights the potential of this traditional medicinal herb as a miRNA-ROS modulating agent in hematologic malignancies, although its effect on exosomal pathways remains unexplored [151]. Icariin, a flavonoid glycoside derived from *Epimedium* species, suppresses proliferation and induces apoptosis in ovarian cancer A2780 cells by downregulating oncogenic miR-21 and upregulating its targets PTEN and reversion-inducing cysteine-rich protein with Kazal motifs (RECK). The study highlights the miR-21/PTEN/RECK axis as a key mechanism underlying its anticancer effect, although exosome involvement has not yet been explored [152]. Qizhu Jianwei decoction (QZJWD), a traditional Chinese medicine formula, was shown to promote ferroptosis in gastric cancer by increasing ROS and lipid peroxidation markers while reducing antioxidant enzyme activity. Mechanistically, QZJWD downregulated exosomal miR-199–3p, relieving repression of ACSL4 and thereby enhancing ferroptotic cell death in both in vitro and in vivo models [153]. Resveratrol, a polyphenolic compound found in *Vitis vinifera* and *Polygonum cuspidatum*, modulates various miRNAs such as miR-21, miR-34a, miR-145, and miR-200c to induce apoptosis, suppress tumor growth, and regulate oxidative stress. Its effects are mediated through the activation of silent mating type information regulation 2 homolog 1 (SIRT1), p53, and AMP-activated protein kinase (AMPK) signaling, though its role in exosome biology remains unexplored and merits further investigation [154]. Shikonin, a natural naphthoquinone from *Lithospermum erythrorhizon*, induces necroptosis in chronic myeloid leukemia (CML) cells by downregulating miR-92a-1-5p and activating the RIPK1/RIPK3/MLKL pathway. This miRNA modulates oxidative stress and necroptosis by targeting MLKL, and its inhibition significantly suppressed tumor growth in vivo, even in TKI-resistant CML models. These findings suggest miR-92a-1-5p as a novel target for overcoming drug resistance in CML via ROS-associated mechanisms [155]. *Spatholobus suberectus Dunn* (SSD) induces apoptosis in myeloma and leukemia cells by increasing reactive oxygen species (ROS) levels and triggering ER stress-related signaling via activating transcription factor 2 (ATF2) and CHOP. SSD also significantly downregulates miR-657, an onco-miRNA, and its effects were reversed by miR-657 mimics and ROS scavengers, highlighting its potential as a redox-targeting therapeutic agent [156]. Quercetin, a flavonol abundant in fruits and vegetables, exhibits potent anticancer activity by modulating ROS levels in cancer and cancer stem cells. It exerts dual effects, scavenging excessive ROS to protect normal cells while inducing ROS-mediated apoptosis in malignant cells through pathways involving AMPK, p53, MAPK, and mitochondrial dysfunction. Although extensively studied in redox signaling, its role in exosomal communication remains to be explored [157]. Wogonin, a flavonoid derived from *Scutellaria baicalensis*, has demonstrated significant antitumor effects in gastric cancer cells by inducing cell cycle arrest and apoptosis via JAK-STAT3 inhibition. The study also suggests wogonin’s potential modulation of ROS levels and post-transcriptional regulation through miRNAs such as miR-155 and miR-145, though its exosome-related effects remain unexplored [158]. Collectively, these findings underscore the therapeutic potential of natural compounds as modulators of the ROS–miRNA–exosome axis across diverse cancer models. While many phytochemicals demonstrate potent regulation of redox balance and miRNA signaling, their role in exosome biogenesis and cargo modulation remains largely unexplored. This section emphasizes the need for future research to bridge this knowledge gap by systematically investigating how herbal medicines and their bioactive constituents may modulate exosome-mediated communication, ultimately enabling the development of redox-targeted, miRNA-guided cancer therapeutics grounded in natural product pharmacology.

However, it should be noted that the majority of findings discussed in this section are based on preclinical or in vitro studies. Thus, while the mechanistic insights are promising, their translational relevance requires further validation through rigorous in vivo and clinical research. Furthermore, while ROS modulation is a common denominator among these compounds, the magnitude and direction of ROS effects are highly dependent on compound-specific pharmacokinetics, experimental models, and delivery contexts. For this reason, we intentionally avoided a concentration-based or numerical ROS ranking, particularly in the case of multi-component herbal extracts, where standardization is technically infeasible. Instead, we prioritized mechanistic relevance and miRNA–exosome integration, while noting general ROS modulation trends for single-compound studies.

## 4. Clinical Implications and Therapeutic Applications

Exosome-mediated delivery of miRNAs has emerged as a promising therapeutic approach due to their ability to modulate reactive oxygen species (ROS)-related signaling pathways in diverse disease contexts. Recent findings have shed light on specific miRNA–ROS interactions within exosomal systems that hold clinical relevance across oncology and fibrosis. In head and HNSCC, Zhang et al. demonstrated that exosomal miR-9-5p derived from HPV-positive tumor cells significantly downregulated TGF-β receptor 2 (TGFBR2) and NOX4, resulting in reduced ROS levels and inhibition of CAF differentiation. This study revealed that exosomal miR-9-5p modulates redox balance in the tumor microenvironment, impairing stromal remodeling and aggressiveness of HNSCC [159]. Separately, Zhou et al. reported that Let-7-enriched exosomes from menstrual blood-derived mesenchymal stem cells (MenSCs) attenuate ROS-induced epithelial damage and mitochondrial DNA release in idiopathic pulmonary fibrosis (IPF). Let-7 targets LOX-1 to suppress the NLRP3 inflammasome, reducing downstream pyroptosis and inflammatory fibrosis. This study highlights a novel antioxidant and anti-fibrotic mechanism by which MenSC-derived exosomes mitigate oxidative injury [160]. Moreover, Chen et al. demonstrated that natural exosome-like nanovesicles derived from edible tea flowers (TFENs) can effectively suppress metastatic breast cancer via ROS amplification and gut microbiota modulation [161]. These plant-derived vesicles exhibited strong intracellular uptake and selectively induced oxidative stress in cancer cells, leading to mitochondrial damage, cell cycle arrest, and apoptosis. Notably, TFENs accumulated in both primary breast tumors and lung metastatic sites, showing comparable therapeutic effects via intravenous and oral administration. This study not only highlights the therapeutic versatility of edible plant-derived nanovesicles but also suggests their potential clinical applicability as a green, safe, and scalable exosomal platform that modulates the tumor microenvironment through ROS and host microbiota regulation. Together, these studies demonstrate the translational potential of engineered or naturally enriched exosomes as targeted therapies. By selectively packaging miRNAs or bioactive compounds that modulate redox signaling and immune activation, exosome-based platforms could be customized for precision therapy in cancer and chronic fibrotic diseases. Furthermore, clinical application of natural compounds such as ginsenoside Rg3 and matrine, which enhance dendritic cell-mediated antitumor immunity via ROS-dependent exosomal communication, offers new perspectives in the immunotherapeutic landscape [112]. While exosome-based strategies offer compelling therapeutic avenues, current clinical translation is limited by challenges such as scalability, cargo loading efficiency, and biological variability. Furthermore, most studies cited herein propose therapeutic implications based on early-phase or preclinical investigations, necessitating cautious interpretation regarding their direct clinical applicability. This section underscores how insights from redox biology and exosomal miRNA regulation can be integrated to inform next-generation therapeutic strategies that harness natural cellular communication for disease modulation.

### Limitations and Future Perspectives

Despite significant progress, several limitations remain in the study and application of the exosome–ROS–miRNA axis. Technically, standardization of exosome isolation, purification, and characterization remains a major hurdle, leading to variability across studies. Moreover, accurately quantifying ROS levels and redox dynamics in vivo is challenging due to the short half-life and reactivity of ROS species. From a biological perspective, the heterogeneity of exosomes, both in terms of cell of origin and cargo content, adds complexity to understanding their specific roles. Additionally, the mechanisms governing selective miRNA loading and ROS-related exosome release remain incompletely defined. Further mechanistic studies are needed to dissect the upstream regulators and downstream consequences of these processes. Future research should focus on integrating multi-omics datasets (proteomics, metabolomics, spatial transcriptomics) to gain a systems-level understanding of the exosome–ROS–miRNA network. Clinical translation will require robust biomarkers, scalable exosome engineering platforms, and precise delivery systems. Ultimately, bridging mechanistic insights with clinical applications will pave the way for innovative, redox-informed cancer diagnostics and therapies. Taken together, these limitations emphasize the urgent need for mechanistic clarity, technological standardization, and translational integration to fully harness the therapeutic potential of the ROS–miRNA–exosome triad. By delineating these challenges, this review aims to guide future research toward the strategic development of redox-sensitive, miRNA-driven exosome-based interventions in cancer and fibrosis.

## 5. Discussion

The intricate relationship between ROS, exosomal cargo loading, and miRNA-mediated signaling in the TME represents a rapidly evolving field of study. Our review integrates current knowledge on how oxidative stress not only stimulates exosome biogenesis but also influences the selective sorting of redox-sensitive miRNAs (miR-21, miR-155, and miR-210) through RNA-binding proteins such as hnRNPA2B1 and SYNCRIP [162]. Additionally, cellular players like TAMs, CAFs, neutrophils, and DCs actively contribute to ROS generation and produce exosomes that carry immune-regulatory and pro-tumorigenic signals, further exacerbating disease progression [163]. The NOX1–ROS–exosome axis, in particular, has emerged as a key mechanism driving M2 macrophage polarization, CD8^+^ T cell suppression, and angiogenesis in the TME [164]. Our synthesis also highlights hypoxia as a powerful modulator of this interplay. Exosomal miRNAs derived from hypoxic cancer cells, such as miR-210, miR-135, and others, are not only upregulated but also shown to enhance angiogenesis, endothelial remodeling, and autophagy in recipient cells [165]. This hypoxia-driven exosomal release is often HIF-1α-dependent and occurs alongside acidic pH–mediated exosome uptake, creating a highly efficient communication network for tumor survival and immune evasion [166]. The addition of single-cell transcriptomics and spatial functional genomics has further refined our understanding of these pathways, revealing cell-type-specific miRNA-exosome-ROS signatures across the immune and stromal compartments. While exosome-mediated therapy remains in early stages, studies included in this review clearly demonstrate the therapeutic potential of targeting the ROS–miRNA–exosome axis, particularly by using natural compounds, engineered EVs, or leveraging endogenous mechanisms (MenSC-derived Let-7 or BM-MSCs releasing miR-125a) [167]. Importantly, our review also emphasizes the emerging role of herbal medicines and phytochemicals as promising modulators of the ROS–miRNA–exosome axis. Natural compounds such as Baicalin [168], Berberine, Bu-Shen-Ning-Xin Decoction [145], BK002 [144,169], *Cnidium officinale* [146], curcumin [147], DanShen Decoction [148], *Daemonorops draco* Blume [149], Ginsenoside Rg3 [150], Icariin [170], *Leonurus japonicus* Houttuyn [151], Qizhu Jianwei decoction [153], Quercetin [171], Resveratrol [172], Shikonin [173], *Spatholobus suberectus* [156], and Wogonin [174] have demonstrated the capacity to modulate oxidative stress, regulate miRNA expression, and in some cases, influence exosome secretion or cargo loading. Importantly, recent evidence highlights the potential of natural compounds such as ginsenoside Rg3 and matrine in modulating dendritic cell function via ROS and exosomal pathways. By inducing CRT and HSP release or enhancing co-stimulatory molecule expression (TNF-α, IL-12, CD80, CD86), these agents reshape T cell immunity and complement the established CAF-centered frameworks of ROS–miRNA–exosome regulation [112]. These plant-derived bioactives represent a rich and largely untapped source of novel therapeutic agents that align with the multifaceted regulation observed in cancer biology. Thus, this review not only summarizes the current scientific understanding but also provides a strategic framework for the development of new therapeutics based on herbal medicine. By elucidating the mechanisms by which natural compounds influence key redox and epigenetic pathways, we aim to guide future efforts in drug discovery. In particular, we propose that selectively targeting miRNA mechanisms already validated by various herbal compounds may offer a new horizon in oncology. The miRNA-focused approach enables specific, gene-level modulation of the tumor microenvironment, allowing precision-driven therapy design. This perspective supports the integration of molecular, cellular, and systemic strategies for next-generation cancer management. Ultimately, we contend that cancer’s adaptability stems from its dynamic interaction with its microenvironment, including redox balance, immune evasion, and metabolic reprogramming. Without addressing this ever-evolving interface, therapeutic control of cancer remains incomplete. We hope this review contributes to reframing cancer not merely as a genetic disease, but as an ecological and environmental pathology requiring holistic, multi-targeted strategies. Our aim is to inspire further research and clinical innovation along the ROS–miRNA–exosome axis, particularly through the application of natural compound-based therapeutics.

## 6. Conclusions

This review underscores the central role of ROS in regulating exosome formation, miRNA cargo selection, and intercellular signaling in the tumor microenvironment. The crosstalk between oxidative stress and miRNA trafficking via exosomes not only promotes tumor progression and immune suppression but also opens novel therapeutic avenues. ROS promotes selective miRNA loading into exosomes through redox-sensitive RNA-binding proteins. Hypoxia-induced exosomal miRNAs (miR-210, miR-135) facilitate angiogenesis and immune evasion. Exosomal miRNAs such as miR-9-5p and Let-7 demonstrate promising anti-inflammatory and antifibrotic effects by targeting ROS-generating enzymes and inflammatory signaling. Natural compounds and stem cell-derived exosomes can modulate this axis and offer new avenues for therapeutic development. Of particular importance, this review highlights that the herbal medicines and phytochemicals that modulate ROS levels, miRNA expression, and exosome dynamics represent a promising and underexplored category of cancer therapeutics. By systematically evaluating their molecular targets and interactions within the ROS–miRNA–exosome network, we provide a conceptual basis for future drug development rooted in natural product research. Future directions should include the clinical translation of exosome-based delivery systems, the development of miRNA loading enhancement technologies, and in-depth analysis through single-cell spatial omics to better characterize ROS–miRNA dynamics in vivo. Ultimately, we hope this work serves as a platform for conceptual innovation and therapeutic design, encouraging researchers and clinicians to adopt an integrative, microenvironment-focused lens in the ongoing fight against cancer.

## Figures and Tables

**Figure 1 antioxidants-14-00501-f001:**
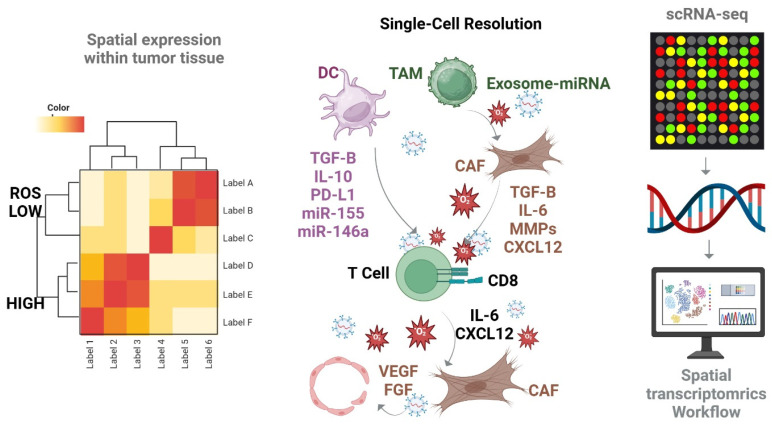
Single-cell and spatial transcriptomic mapping of exosome–ROS–miRNA dynamics in the tumor microenvironment. Spatial transcriptomics combined with single-cell RNA sequencing (scRNA-seq) enables high-resolution mapping of exosome-mediated miRNA exchange and redox signaling within tumor compartments. This process involves key interactions between dendritic cells (DCs), tumor-associated macrophages (TAMs), cancer-associated fibroblasts (CAFs), and T cells, through the transfer of immunoregulatory cytokines (TGF-β, IL-6, IL-10), immune checkpoint proteins (PD-L1), and redox-sensitive miRNAs (e.g., miR-155, miR-146a). These interactions contribute to the suppression of CD8+ T cells, extracellular matrix (ECM) remodeling, and angiogenesis via vascular endothelial growth factor (VEGF) and fibroblast growth factor (FGF). The heatmap (left) illustrates spatial expression patterns of redox gene signatures across tumor zones with varying levels of reactive oxygen species (ROS). The integrated scRNA-seq and spatial transcriptomic workflow (right) provides a comprehensive approach to mapping exosome-mediated cell–cell crosstalk at the tissue level. Abbreviations: cancer-associated fibroblast (CAF), cluster of differentiation 8 (CD8), C-X-C motif chemokine ligand 12 (CXCL12), dendritic cell (DC), Fibroblast Growth Factor (FGF), hypoxia-inducible factor 1-alpha (HIF-1α). Interleukin-6 (IL-6), Matrix Metalloproteinases (MMPs), microRNA (miRNA), Programmed Death-Ligand 1(PD-L1), reactive oxygen species (ROS), single-cell RNA sequencing (scRNA), tumor-associated macrophage (TAM), transforming growth factor beta (TGF-β), Vascular Endothelial Growth Factor (VEGF).

**Figure 2 antioxidants-14-00501-f002:**
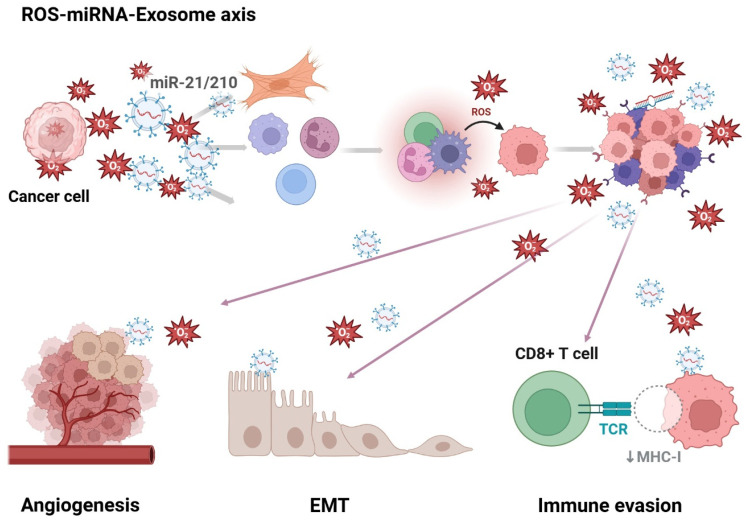
The ROS–miRNA–exosome axis in the tumor microenvironment. Schematic representation of the ROS–miRNA–exosome axis in the tumor microenvironment. Cancer cells generate excessive reactive oxygen species (ROS), which in turn promote the biogenesis and selective miRNA loading of exosomes. These exosomal miRNAs are delivered to surrounding immune and stromal cells, modulating the tumor microenvironment by promoting angiogenesis, epithelial–mesenchymal transition (EMT), and immune evasion. ROS and exosomal miRNA signaling together create a feedback loop that reinforces tumor aggressiveness and therapeutic resistance. Abbreviations: reactive oxygen species (ROS), microRNA (miRNA), epithelial–mesenchymal transition (EMT), T Cell Receptor (TCR), major histocompatibility complex class I (MHC-I). ↓ means decrease.

**Table 1 antioxidants-14-00501-t001:** Cell-type-specific roles in the ROS–miRNA–Exosome axis of the tumor microenvironment.

Cell Type	ROS Role	Key miRNA(s)	Exosomal Impact	Representative Pathway	Reference
CAFs	↑ ROS	miR-522	Inhibit ferroptosis via exosomal transfer	TGF-β/SMAD, NLRP3 inflammasome	[107]
DCs	Moderate ROS producers	miR-155	Enhanced antigen presentation, enhanced T cell activation	CRT, HSP, TNF-α, IL-12, CD80, CD86	[112]
			Suppress antigen presentation via exosomes	IL-10, MHC-II	[107]
TANs	↑ ROS via NOX1	miR-210, miR-451a	Promote autophagy and N2 polarization	HMGB1/TLR4/NF-κB	[105]
TAMs	↑ ROS via NOX2	miR-21, miR-155	Drive M2 polarization and immunosuppression	JAK/STAT3, NF-κB	[84,110,111]
Tumor cells	↑ ROS via metabolism and hypoxia	miR-135, miR-210, miR-23a-3p	Regulate angiogenesis and ferroptosis	HIF-1α, Nrf2/Keap1, GPX4	[113]

Abbreviations: cancer-associated fibroblast (CAF), dendritic cell (DC), glutathione peroxidase 4 (GPX4), heat-shock proteins (HSPs), hypoxia-inducible factor 1-alpha (HIF-1α), high mobility group box 1 (HMGB1), Interleukin-10 (IL-10), Interleukin-12 (IL-12), Janus kinase (JAK), major histocompatibility complex class II (MHC-II), microRNA (miRNA), nuclear factor kappa-light-chain-enhancer of activated B cells (NF-κB), NADPH oxidase isoform ½ (NOX1/NOX2), NOD-like receptor protein 3 (NLRP3), Reactive oxygen species (ROS), signal transducer and activator of transcription 3 (STAT3), tumor-associated macrophage (TAM), tumor-associated neutrophil (TAN), transforming growth factor beta (TGF-β), tumor necrosis factor alpha (TNF-α). ↑ means increase.

**Table 2 antioxidants-14-00501-t002:** Natural compounds targeting the ROS–miRNA–exosome axis: phytochemicals with potential for redox-based therapeutic modulation.

Compound/Extract	Source Plant	Target miRNA	ROS Efficacy	Exosome Study	Reference
*Astragaloside* IV + Ferulic Acid	*Astragalus membranaceus* + *Angelica sinensis*	miR-29b	↓ ROS, ↑ SOD, ↓ MDA via Nrf2 and TGF-β1/Smad3	Not studied	[141]
Baicalin	*Scutellaria baicalensis*	miR-338-3p, let-7c	↑ ROS-mediated apoptosis, ↓ NF-κB, ↓ Bcl-2	Not studied	[142]
Berberine	*Coptis chinensis*	miR-21	↑ ROS, ↓ Bcl-2, ↑ apoptosis	Not Studied	[143]
BK002	Achyranthes japonica + Melandrium firmum	miR-192-5p	↑ ROS-mediated apoptosis via miRNA modulation	Not studied	[144]
Bu-Shen-Ning-Xin Decoction (BSNXD)	Multi-herbal formula	miR-760-3p (via circRNA_012284)	↓ ROS in ovarian granulosa cells via HBEGF modulation	Not Studied	[145]
*Cnidium officinale Makino* (COM)	*Cnidium officinale* Makino	miR-211	↑ ROS→ ↑ER stress-mediated Apoptosis	Not studied	[146]
Curcumin	*Curcuma longa*	miR-34a, miR-34b/c	↑ ROS → activates NRF2 → ↑ miR-34a/b/c	Not studied	[147]
DanShen Decoction (DSD)	*Salvia miltiorrhiza*, *Santalum album*, *Amomum villosum*	miR-93-5p, miR-15b-5p, miR-16-5p	↓ROS, ↓ Pyroptosis, ↑ SOD	BMSC-derived exosomes	[148]
*Daemonorops draco* (DD)	*Daemonorops draco* Blume	miR-216b	↑ ROS, ↑ Apoptosis	Not studied	[149]
Ginsenoside Rg3	*Panax ginseng*	miR-204	↓ ROS, ↓ EMT and Stemness	Exosomal miR-204	[150]
Icariin	*Epimedium* spp.	miR-21	↓ ROS via PTEN/RECK, ↑ Apoptosis	Not Studied	[152]
*Leonurus japonicus Houttuyn*	*Leonurus japonicus*	miR-19a-3p	↑ ROS → ↑ ER stress-mediated Apoptosis	Not studied	[151]
Qizhu Jianwei decoction (QZJWD)	*Astragalus membranaceus, Polygonatum odoratum, Atractylodis macrocephala, Curcuma phaeocaulis*,	miR-199–3p	↑ ROS, ↑ MDA, ↑ Fe^2+^, ↓ GPx → ↑ Ferroptosis	Exosomal miR-199–3p	[153]
Quercetin	*Various plants*		Dual role depending on dose	Emerging evidence	[157]
Resveratrol	*Vitis vinifera*, *Polygonum cuspidatum*	miR-21, miR-22, miR-34a, miR-145, miR-200c	↓ ROS, anti-aging, ↑ apoptosis	Not studied	[154]
Shikonin	*Lithospermum erythrorhizon*	miR-92a-1-5p	↑ ROS; necroptosis via MLKL pathway	Not studied	[155]
*Spatholobus suberectus Dunn* (SSD)	*Spatholobus suberectus*	miR-657	↑ ROS → ↑ER stress-mediated apoptosis	Not studied	[156]
Wogonin	*Scutellaria baicalensis*	miR-155, miR-145 (indirect)	Slight ↑ ROS, apoptosis induction, mitochondrial dysfunction	Not studied	[158]

Abbreviations: Adenosine Monophosphate-activated protein kinase (AMPK), bone marrow-derived mesenchymal stem cells (BMSCs), cancer stem cells (CSCs), Endoplasmic Reticulum (ER), enhanced permeability and retention (EPR), epithelial–mesenchymal transition (EMT), extracellular vesicle (EV), Ferroptosis Suppressor Protein 1 (GPx), high mobility group box 1 (HMGB1), hypoxia-inducible factor 1-alpha (HIF-1α), Janus kinase 1 (JAK1), malondialdehyde (MDA), Mammalian Target of Rapamycin (mTOR), mesenchymal stem cells (MSCs), mitogen-activated protein kinase (MAPK), mononuclear phagocyte system (MPS), multivesicular bodies (MVBs), nuclear factor erythroid 2–related factor 2 (NRF2), nuclear factor kappa-light-chain-enhancer of activated B cells (NF-κB), NADPH oxidase 1 (NOX1), Natural Killer Cells (NK cells), oxidative stress (OS), Programmed Cell Death Ligand 1 (PD-L1), phosphatase and tensin homolog (PTEN), reactive oxygen species (ROS), reactive oxygen species–mediated cell death (RMCD), reversion-inducing cysteine-rich protein with Kazal motifs (RECK), signal transducer and activator of transcription 3 (STAT3), Superoxide Dismutase (SOD), tumor-associated macrophages (TAMs), tumor-derived exosomes (TEXs), tumor microenvironment (TME), Vascular Endothelial Growth Factor (VEGF). ↑ means increase, ↓ means decrease.

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
