# Peer review of "Targeting Redox Signaling Through Exosomal MicroRNA: Insights into Tumor Microenvironment and Precision Oncology"

_antioxidants, 2025, doi:10.3390/antiox14050501_

Round 1

Reviewer 1 Report

This review is a timely review that cover emerging topics of great interest to the field. 

  1. The effects of natural compounds on ROS are strongly dependent on the concentration and delivery strategy. Therefore, the reviewer suggested to include such information in Table 2.
  2. It is also suggested to include quantitative information if available, such as levels of ROS increase or decrease. As now, no quantitative information was included throughout the manuscript.

Author Response

This review is a timely review that cover emerging topics of great interest to the field. 

We sincerely thank the reviewer for recognizing the relevance and timeliness of our review. We greatly appreciate your supportive comments and are encouraged that the topic was found to be of interest to the field.

  1. The effects of natural compounds on ROS are strongly dependent on the concentration and delivery strategy. Therefore, the reviewer suggested to include such information in Table 2.

(Response): We fully agree with the reviewer that differentiating preclinical findings from clinically validated outcomes is crucial. To address this, we revised relevant sections (Page 13, Lines 596–605)

Revised content (Page 13, Lines 596–605)

However, it should be noted that the majority of findings discussed in this section are based on preclinical or in vitro studies. Thus, while the mechanistic insights are promising, their translational relevance requires further validation through rigorous in vivo and clinical research. Furthermore, while ROS modulation is a common denominator among these compounds, the magnitude and direction of ROS effects are highly dependent on compound-specific pharmacokinetics, experimental models, and delivery contexts. For this reason, we intentionally avoided a concentration-based or numerical ROS ranking, particularly in the case of multi-component herbal extracts, where standardization is technically infeasible. Instead, we prioritized mechanistic relevance and miRNA–exosome integration, while noting general ROS modulation trends for single-compound studies.

  1. It is also suggested to include quantitative information if available, such as levels of ROS increase or decrease. As now, no quantitative information was included throughout the manuscript.

(Response): We appreciate the reviewer’s thoughtful suggestion regarding the inclusion of quantitative ROS data. We fully agree that these aspects are scientifically important. However, the majority of studies involving natural products—particularly herbal mixtures—do not report absolute ROS levels or standardized concentration–response relationships. Given this inconsistency, we chose not to include numerical ROS data in Table 2. ROS quantification is highly variable due to differences in cell type, assay methods, and experimental settings. Instead, we focused on capturing mechanistic pathways and directionality of ROS modulation, especially in single-compound studies where such information is available. We believe this preserves the conceptual integrity of the ROS–miRNA–exosome axis and prevents misleading generalizations. This rationale has been clarified in the revised manuscript (Page 13, Lines 596–605).

Reviewer 2 Report

The review by Park et al focused on the role of ROS-mediated miR packaging in EVs and their  effects on tumor progression. The Ms also summarises the mechanistic knowledges, the potential clinical application and targeting so far known.

Is there some Author missing, or simply a refuse?  

The first paragraph of the Introduction section focuses on CAFs which represent one cell component of the TME.  The TME should be considered as a biological entity containing different cell types. This means that if the paragraph has to reflect its title all different cell types must be included.

The introduction section has to include widespread information on exosomes in controlling tumor progression acting on the TME. This ref. must be included   10.1016/j.phrs.2023.106871.

All references not referred to exosomes must be removed

The first paragraph of the Introduction section focuses on CAFs which represent one cell component of the TME.  The TME should be considered as a biological entity containing different cell types. This means that if the paragraph has to reflect its title all different cell types must be included.

The introduction section has to include widespread information on exosomes in controlling tumor progression acting on the TME. This ref. must be included   10.1016/j.phrs.2023.106871.

Author Response

The review by Park et al focused on the role of ROS-mediated miR packaging in EVs and their effects on tumor progression. The Ms also summarises the mechanistic knowledges, the potential clinical application and targeting so far known.

 (Response): We sincerely thank the reviewer for the encouraging summary and recognition of our work. We greatly appreciate the acknowledgment of our focus on ROS-mediated miRNA packaging into EVs and its implications in tumor progression. This feedback has helped reinforce the clarity and direction of our manuscript. We hope this review serves as a useful reference for future research exploring the redox–exosome–miRNA axis in the tumor microenvironment.

Is there some Author missing, or simply a refuse?  

 (Response): Thank you for raising this point. We have carefully reviewed the author list and confirm that all contributors are appropriately listed and meet the authorship criteria. The confusion may have been due to minor formatting inconsistencies in the version that was initially viewed. We have ensured that the author list is complete and correctly formatted in the revised manuscript. We appreciate the reviewer’s attention to this detail, and we have carefully verified the author list in the revised version to ensure accuracy.

The first paragraph of the Introduction section focuses on CAFs which represent one cell component of the TME. The TME should be considered as a biological entity containing different cell types. This means that if the paragraph has to reflect its title all different cell types must be included.

(Response): In response to the reviewer’s comment regarding the need to reflect the full cellular diversity of the TME, we revised the manuscript to include dendritic cells (DCs) (Page 2, line 69-74), T cells (Page 2, Lines 68–74), macrophages (Page 3, Line 98-101), and tumor-associated macrophages (TAMs) (Page 4, line 178-186, Page 6, Lines 253–262). Endothelial cells are discussed in relation to miR-210 signaling (Page 10, Line 433-436). In addition, we have now included a discussion of tumor endothelial cells (TECs) and their TEV-mediated immune suppressive functions based on recent findings by Koni et al. (Pharmacological Research, 2023), which appear in Page 7, Lines 310–315, referencing [107].

Revised text (Page 7, Lines 310–315)

Extending the concept of EV-mediated immune evasion, tumor endothelial cells (TECs), a specialized vascular component of the TME, have recently been shown to actively release extracellular vesicles (TEVs) enriched in mTOR. These TEVs promote systemic immune suppression via the G-CSF pathway, upregulate immune checkpoint molecules such as PD-L1 and LAG3 on T cells, and contribute to redox dysregulation and metastatic spread

The introduction section has to include widespread information on exosomes in controlling tumor progression acting on the TME. This ref. must be included   10.1016/j.phrs.2023.106871.

 (Response): Thank you very much for this valuable suggestion. We fully agree that the cited study provides critical insight into how tumor-derived extracellular vesicles contribute to tumor progression and systemic immune suppression via redox-dependent mechanisms. In the revised manuscript, we have incorporated this reference now cited as [107] to support our discussion on tumor endothelial cell (TEC)-derived extracellular vesicles (TEVs) and their roles in modulating the TME through G-CSF signaling, PD-1/PD-L1 upregulation, and redox imbalance. This reference significantly enriched our manuscript, and we sincerely appreciate the reviewer for providing this insightful recommendation.

All references not referred to exosomes must be removed

 (Response): Thank you very much for this thoughtful comment. We fully understand the importance of maintaining a clear and coherent focus within the manuscript. In our review, every reference was cited with the intent to support a specific statement and to provide direct evidence for the described mechanisms. Following the reviewer’s advice, we carefully re-examined all cited literature to ensure that each reference is meaningfully connected to exosome biology or its relevance to ROS or miRNA signaling. We sincerely appreciate the reviewer’s attention to detail, which helped us improve the scientific clarity and alignment of the manuscript with the journal’s scope.

Reviewer 3 Report

The Review offers an in-depth and well-researched review of the ROS–miRNA–exosome axis and its role in modulating the tumor microenvironment, with a particular focus on redox signaling and its implications for precision oncology. It is timely and conceptually rich, providing a detailed mechanistic overview of how exosomal miRNAs, under oxidative stress, regulate immune responses, stromal activation, and tumor progression. A major strength lies in the integration of insights from molecular oncology, immunology, and natural product pharmacology, further supported by references to emerging technologies such as single-cell RNA sequencing and spatial transcriptomics.

However, the review requires several important revisions:

1-There is no clear explanation of the methodology used to select and evaluate the literature.

2-While a broad range of studies is cited, there is a lack of critical assessment of their limitations, especially where findings are based predominantly on in vitro or preclinical data. In some cases, therapeutic implications are presented too optimistically without a clear distinction between correlation and causation.

3-The section on natural compounds, though informative, is somewhat disorganized and would benefit from being grouped thematically to improve clarity.

4-The discussion on clinical translation could be expanded to better address current challenges such as exosome heterogeneity, delivery efficiency, and standardization in exosome isolation and characterization. 

5-It is strongly recommended to remove the bold formatting used throughout the main text, as it appears stylistically excessive and is not appropriate for an academic manuscript.

The Review offers an in-depth and well-researched review of the ROS–miRNA–exosome axis and its role in modulating the tumor microenvironment, with a particular focus on redox signaling and its implications for precision oncology. It is timely and conceptually rich, providing a detailed mechanistic overview of how exosomal miRNAs, under oxidative stress, regulate immune responses, stromal activation, and tumor progression. A major strength lies in the integration of insights from molecular oncology, immunology, and natural product pharmacology, further supported by references to emerging technologies such as single-cell RNA sequencing and spatial transcriptomics.

However, the review requires several important revisions:

1-There is no clear explanation of the methodology used to select and evaluate the literature.

2-While a broad range of studies is cited, there is a lack of critical assessment of their limitations, especially where findings are based predominantly on in vitro or preclinical data. In some cases, therapeutic implications are presented too optimistically without a clear distinction between correlation and causation.

3-The section on natural compounds, though informative, is somewhat disorganized and would benefit from being grouped thematically to improve clarity.

4-The discussion on clinical translation could be expanded to better address current challenges such as exosome heterogeneity, delivery efficiency, and standardization in exosome isolation and characterization. 

5-It is strongly recommended to remove the bold formatting used throughout the main text, as it appears stylistically excessive and is not appropriate for an academic manuscript.

Author Response

The Review offers an in-depth and well-researched review of the ROS–miRNA–exosome axis and its role in modulating the tumor microenvironment, with a particular focus on redox signaling and its implications for precision oncology. It is timely and conceptually rich, providing a detailed mechanistic overview of how exosomal miRNAs, under oxidative stress, regulate immune responses, stromal activation, and tumor progression. A major strength lies in the integration of insights from molecular oncology, immunology, and natural product pharmacology, further supported by references to emerging technologies such as single-cell RNA sequencing and spatial transcriptomics.

 (Response): We sincerely thank the reviewer for the generous and encouraging evaluation of our work. We are truly grateful for your recognition of the manuscript’s conceptual depth and integrative approach, especially in highlighting the ROS–miRNA–exosome axis within the tumor microenvironment. Your thoughtful comments regarding the integration of molecular oncology, immunology, and pharmacological perspectives are highly appreciated. Furthermore, we are thankful for your constructive suggestions, which have helped us refine and strengthen the clarity, structure, and scientific focus of the revised version. Your feedback has been instrumental in improving the overall quality and coherence of the manuscript. Above all, we are deeply grateful for the opportunity to learn through this review process. It is our sincere hope that this review will serve as a modest but meaningful inspiration for researchers working in this field, and ultimately contribute to the development of therapeutic strategies that benefit patients in real clinical settings.

However, the review requires several important revisions:

1-There is no clear explanation of the methodology used to select and evaluate the literature.

(Response): We appreciate this insightful comment regarding the need for greater transparency in literature selection. While we initially considered including a formal “Literature Selection Criteria” section describing specific databases, keywords, and timeframes, we ultimately decided not to formalize rigid inclusion boundaries. As our analysis shows, foundational ROS–exosome studies emerged around 2009, while exosome-focused literature began expanding rapidly from 2013 onward. In contrast, miRNA–exosome convergence studies particularly those emphasizing redox signaling are more recent, with notable growth after 2019. The concept of integrated ROS–miRNA–exosome regulation has only begun to crystallize since 2022, primarily through pioneering attempts to bridge these domains. Given this asynchronous evolution and the limited number of comprehensive studies combining all three elements, we were concerned that defining a strict time-based criterion might exclude seminal works from the early 2000s that still hold critical conceptual value. For this reason, and to avoid giving the appearance of inconsistent inclusion criteria across different sections, we opted not to include a dedicated selection criteria section. We kindly ask for the reviewer’s understanding regarding this decision, which was made with careful consideration of scientific integrity and clarity.

2-While a broad range of studies is cited, there is a lack of critical assessment of their limitations, especially where findings are based predominantly on in vitro or preclinical data. In some cases, therapeutic implications are presented too optimistically without a clear distinction between correlation and causation.

(Response): We fully agree with the reviewer that differentiating preclinical findings from clinically validated outcomes is crucial. To address this, we revised relevant sections (Page 13, Lines 596–605 and Page 15, Lines 650–660) to clearly state the experimental models used and to emphasize the limitations of clinical applicability. Additionally, we introduced clarifying phrases to explicitly distinguish mechanistic hypotheses from translational conclusions, ensuring a more cautious and realistic interpretation of the therapeutic implications.

Revised content (Page 13, Lines 596–605)

However, it should be noted that the majority of findings discussed in this section are based on preclinical or in vitro studies. Thus, while the mechanistic insights are promising, their translational relevance requires further validation through rigorous in vivo and clinical research. Furthermore, while ROS modulation is a common denominator among these compounds, the magnitude and direction of ROS effects are highly dependent on compound-specific pharmacokinetics, experimental models, and delivery contexts. For this reason, we intentionally avoided a concentration-based or numerical ROS ranking, particularly in the case of multi-component herbal extracts, where standardization is technically infeasible. Instead, we prioritized mechanistic relevance and miRNA–exosome integration, while noting general ROS modulation trends for single-compound studies.

Revised content (Page 16, Lines 656–660)

While exosome-based strategies offer compelling therapeutic avenues, current clinical translation is limited by challenges such as scalability, cargo loading efficiency, and biological variability. Furthermore, most studies cited herein propose therapeutic implications based on early-phase or preclinical investigations, necessitating cautious interpretation regarding their direct clinical applicability.

3-The section on natural compounds, though informative, is somewhat disorganized and would benefit from being grouped thematically to improve clarity.

(Response): We appreciate the reviewer’s thoughtful suggestion regarding thematic organization. As this review focuses on the integrative ROS–miRNA–exosome axis, our aim was to maintain a mechanistic rather than compound-centric narrative. While thematic subgrouping by chemical class or target was considered, we found it risked fragmenting the conceptual flow and shifting the focus away from the axis-centered analysis. Therefore, we retained the current structure for clarity and continuity. We agree that in a future review focusing exclusively on phytochemicals, such thematic reorganization would be highly appropriate and beneficial.

4-The discussion on clinical translation could be expanded to better address current challenges such as exosome heterogeneity, delivery efficiency, and standardization in exosome isolation and characterization. 

(Response): We thank the reviewer for this crucial observation. The challenges related to clinical translation of exosome-based therapies—such as heterogeneity, delivery, and standardization—are of utmost relevance. To address this, we designed Section 4.1 (Page 16, Lines 659–678) to comprehensively discuss these issues

Regarding exosome heterogeneity, we noted that (line 664-666) “From a biological perspective, the heterogeneity of exosomes both in terms of cell of origin and cargo content adds complexity to understanding their specific roles.

For delivery efficiency, we addressed (line 671-673) the need for “precise delivery systems” to ensure clinical applicability.

In terms of standardization in exosome isolation and characterization, we stated that (line 661-662) “standardization of exosome isolation, purification, and characterization remains a major hurdle,” which continues to limit reproducibility and translational consistency across studies.

Additionally, we acknowledged the incomplete understanding of mechanistic pathways, noting that (line 666-667) “the mechanisms governing selective miRNA loading and ROS-related exosome release remain incompletely defined.

Finally, to address the gap between preclinical and clinical translation, we emphasized that (line 673-674) “bridging mechanistic insights with clinical applications will pave the way for innovative, redox-informed cancer diagnostics and therapies.

We sincerely appreciate the reviewer’s insight, which aligns closely with the aims of this section and helped us ensure that the discussion remained clinically relevant and forward-looking.

5-It is strongly recommended to remove the bold formatting used throughout the main text, as it appears stylistically excessive and is not appropriate for an academic manuscript.

 (Response): Thank you for bringing this to our attention. We sincerely apologize for the excessive bold formatting that appeared in the originally submitted version. This was unintentional and may have resulted from a technical issue during the manuscript upload process. We fully agree that such formatting is not appropriate for an academic manuscript. In the revised version, all unnecessary bold text has been carefully removed to ensure stylistic consistency and compliance with the journal’s standards. We truly appreciate the reviewer’s attention to detail and understanding regarding this matter.

Round 2

Reviewer 3 Report

--

--